# Multiple Vulnerabilities in Medical Settings: Invisible Suffering of Doctors

**Daria Litvina \*** [ID]**, Anastasia Novkunskaya and Anna Temkina**

Faculty of Sociology and Philosophy, Gender Studies Program, European University at Saint-Petersburg, 191187 Saint-Petersburg, Russia; anovkunskaya@eu.spb.ru (A.N.); atemkina@gmail.com (A.T.)
\* Correspondence: litvina.darya@gmail.com

**Abstract:** While there is a substantive amount of literature on vulnerability of different kinds of patients in different settings, medical professionals are usually considered as the ones who possess power and gain a privileged position. In this paper, we aim to demonstrate that in a certain context physicians—a social group which is usually referred to as "powerful"—consider themselves vulnerable, and this positioning may influence patients in turn. This perspective highlights the complexity of interactions within medical organizations and contributes to the studies of sensitive topics and vulnerable groups. We conceptualize vulnerability of doctors and discuss what can be problematic in powerful doctors' position. We describe some features of the post-Soviet context of Russian healthcare system and maternity care, both of which can be conceptualized as a hybrid of legacy of Soviet paternalism and new neoliberal reforms, managerialism and marketization. Empirical research is based on the ethnographic evidence from the study of a Russian perinatal center. In this article, we explore specific "existential" and "moral" vulnerabilities of medical professionals who routinely have to cope with multiple challenges, such as complicated clinical tasks, rigid control of different state bodies and emotional responses of suffering patients. We argue that there is a bond between the vulnerability of doctors and that of patients, whose position becomes more problematic as professionals become more vulnerable. At the end, we discuss methodological and theoretical implications of our research.

**Keywords:** vulnerability; maternity care; healthcare; doctors; perinatal center; suffering

---

## 1. Introduction

The goal of this paper is to examine an invisible vulnerability of doctors, whose power is usually taken for granted by social researchers. We examine their vulnerability in the context of perinatal center—one of specialized high-tech maternity care units in Russia. Vulnerability in social sciences is frequently interpreted as a one-sided process within binary relationships: since doctors have a ruling position, professional knowledge and agency, it is patients who are powerless and suffering. The vulnerability of medical professionals is rarely discussed in studies of vulnerable groups and sensitive experiences.

The term "vulnerable" is a concept that sometimes is used interchangeably with such terms as "sensitive", "hard to reach" and "hidden populations" [1] (p. 3). Vulnerability is defined as a lack of autonomy and independence, bodily and psychological insecurity, marginalized or deviant status, lack of acknowledgement within the society [1]. This term refers to individuals and social groups, as well as to certain situations and topics. Researchers have been studying vulnerability in connection to taboo topics that are emotionally overwhelming [2] (p. 6)—the ones concerning intimate, discrediting or incriminating experience [2], such as death, grief, violence, AIDS, drugs and homelessness. Vulnerable

groups are exposed to discrimination, intolerant attitude, subordination. In particular, they include people who have certain health-related conditions, such as terminally ill or mentally ill [3].

Doctors are rarely characterized as a vulnerable group, but within certain circumstances, they can be recognized as "vulnerable". However, based on analysis of the post-Soviet maternity care and inductive analysis of empirical data, we argue that Russian doctors could systematically experience vulnerability and that different kinds of vulnerabilities of doctors and patients are interwoven. Our analysis deals with social and institutional (rather than psychological) dimensions of doctors' vulnerabilities. Sociological discussion on vulnerability in medical settings is the starting point of our research. Vulnerability is usually seen as an inherent quality of certain social groups (but not others), while in our approach it has many dimensions and might be attributed to relatively "powerful" groups.

Our research is aimed at examining social arrangements of interactions in medical organization, feelings of its participants and barriers for patient-centered approach to maternity care in Russia. Doctors in Russia have to satisfy contradictory clinical, bureaucratic and social requirements. The social position of medical professionals is characterized by lack of autonomy and high level of subordination. Their positioning is contextualized by such processes as hybridization of market, contemporary managerial reforms and the legacy of soviet paternalism. 'Unjust' (from doctors' point of view) demands from patients, management and authorities; routine collisions with severe clinical conditions; emotional situations and absence of various resources makes doctors vulnerable in special ways, which we define as "existential vulnerability" and "moral vulnerability".

The structure of this article is as follows. First, we describe data and method. After that, in background section, we conceptualize vulnerability of doctors and discuss certain problematic issues related to doctors' powerful position. Then, we describe some features of the post-Soviet context of Russian healthcare system and maternity care in particular. Perinatal center is considered as a special case. Following empirical sections are based on the ethnographic evidence from the study of a perinatal center. We introduce the analytical terms "existential vulnerability" and "moral vulnerability", which were inductively constructed to explore multiple challenges which medical professionals routinely have to cope with. Then, we argue that there is a connection between vulnerability of doctors and that of patients, whose position becomes more problematic as professionals become more vulnerable. In the end we discuss methodological and theoretical implications of our research, concerning (1) the subject of vulnerability, (2) meaning of the context in exploring vulnerabilities or vulnerable groups, (3) interconnections between vulnerabilities of doctors and those of patients and (4) the position and actions of the researcher in the empirical field when dealing with multiple vulnerabilities.

## 2. Materials and Methods

The aim of this project is to explore various attitudes of medical professionals, patients, and other actors in medical environment in order to identify potential tensions, conflicts and complaints in medical settings and determine the ways to cope with them. We focus on the interactions between medical professionals and patients, as well as between the staff members and different departments of perinatal center.

The research has been built on fundamentals of institutional ethnography developed by Dorothy Smith [4]. According to it, communication (a transmission of information and the ways actors implement it to their work) links local practices with the broader institutional context [4] (p. 169). Adapting the logic of the "extended case method", this methodology allows us to study the connections between macro-structural changes and practices at the micro-level [5,6].

This methodology provides opportunities to observe practices and understand the social meanings and structures, which stand behind them. A comprehensive study of different social perspectives allows us to identify organizational tensions in the Perinatal Center and explain what challenges and at what levels (organizational, interactional) are systematically reproduced.

The empirical base of the study (2019) consists of:

1. 33 sessions of ethnographic observations (including field conversations, field interviews, analysis of material environment and documents) in one of the Russian perinatal centers. The collective of three field researchers conducted 249 hours of observations, which were recorded as 391 pages of field notes.
2. Observation at medical events (including conferences, seminars, trainings) at the research site and in the other medical organizations.
3. Analysis of written complaints by patients.

The results of the current research have also been triangulated with the previously gathered data. We did not include this data into analysis and do not refer to it in this article (as it does not address its main questions and goals), but it contributes to our understanding of the social processes within healthcare system in Russia:

1. Analysis of documents (State laws, orders and projects; online reviews (n = 35) (2018); posts of flashmob "violence in delivery" (#nasilie_v_rodah) (n = 50) (2018))
2. Interviews with patients (n = 10) and healthcare professionals of perinatal center (n = 20) (2018).
3. 16 sessions of non-systematic observations at perinatal center (2018);

In the text we use the term "professionals" interchangeably with 'medical practitioners' to denote doctors of different specializations, nurses and midwifes working in various departments. We mostly focus on doctors—obstetricians, neonatologists, anesthesiologists, pediatricians and others. On one hand, they are the ones who make decisions and take responsibility (both in front of controlling bodies and patients) for negative effects of treatment, birth traumas, lethal outcomes, etc. On the other hand, both in theoretical debate and empirically, they are more associated with power, high status and emotional neutrality in medical institutions than nurses and midwives, who are less powerful and more associated with care and involvement. In this article, we want to show that due to these reasons "powerful" doctors are becoming vulnerable in a very specific way. At the same time, we recognize the significance of nursing staff, who do a lot of emotional labor and faces different challenges, and consider them as vulnerable too.

The research was authorized by the administration of the perinatal center and was approved by the ethical committee of Saint-Petersburg Association of Sociologists (SPAS). All of the participants were informed about the study and were guaranteed confidentiality and anonymity.

## 3. Background Section

### 3.1. Multiple Vulnerabilities in Healthcare

Despite the radical transformations of healthcare within the last decades globally, doctor–patient relationships have been conventionally characterized by asymmetry in terms of power, agency, knowledge and control. This asymmetry goes back both to a normative paternalistic model described by Parsons (1951) [7] and to medical power and medicalization in Foucault's terms [8] and their numerous progenies. It implies a type of doctor–patient relationships, in which the patient seeking medical help performs a "sick role", which undermines his dependence on a doctor, vulnerability, incompetence, and helplessness. While Parsons explained such distribution of power as a functional and mutually beneficial cooperation, his concept has been widely criticized by scholars, who interpreted such relationships rather as conflicting and problematic. As healthcare systems were changing, the social positions of doctors and patients within them were changing too. The critical view of social scientists also shifted from social roles and norms towards interactions, practices and structural limitations. However, the idea of power as a part of medical professions was still a cross-cutting issue for many scholars. One of the classics of sociology of medicine, Eliot Freidson, proposed a conceptual model, in which an attempt to gain control over laymen (as well as to cure them) characterizes medical professionals and their interactions with patients, which means that medical experts' authority and patients' autonomy have been in conflict [9]. References to Foucault are important for interpretation

not only of patients as constructed though medicalization, normalizing medical gaze and power [8] but for understanding of both patients' and doctors' subjectivities as constructed in medical settings and depending on each other [8].

At the moment, one of the most facilitated concepts both in public health and scholarly research is a patient-centered model of medical care, which aims to establish egalitarian relationships between patients and healthcare providers. However, the concept itself is still being discussed [10], and practice, framed by this principle, has to deal with different limitations. Despite certain organizational steps towards patient-centeredness in Russia, basic elements of asymmetry in patient–doctor relationships remain the same as in the paternalistic model. Power and knowledge are still exclusively attributed to professionals, and patients are still positioned in interactions as objects of medical manipulations. Particularly, in the sphere of obstetrics and maternity care, which tends to be the frontier of patient-centered change in a global context, in Russia the notion "doctor knows best" is still quite relevant. According to sociologists and clinicians, women are mostly deprived of the possibility to act, make decisions, withstand the aggressive manipulations from medical personnel [11]. In many researches, a patient turns out to be a powerless and suffering figure.

Vulnerability of patients is evident not only due to their physical suffering but also due to their subordinate social positions and respective emotional experiences. Loss of self is among the main indicators. According to the study conducted by Kathy Charmaz [12], the main suffering of chronically ill people could be described as the "loss of self" [12] (p. 168). As Ian Wilkinson and Arthur Kleinman put it, "The most terrible and disabling events of suffering tend to involve us in the experience of losing our roles and identities" [13] (p. 9). There are multiple ways of overcoming the position of powerlessness for patents discussed in literature. Their subjectivity changes as they receive voice, became storytellers, consumers, citizens [14–16]. Alongside with the fact that patient gets agency through getting voice, neoliberal transformations in healthcare (both globally and in Russia) also contribute to changes of a patient, who becomes not just a passive suffering sick person but an active consumer, who has resources to make choices, to decide and to get actively involved into the process of cure. In maternity care women make choices and become demanding consumers [17].

By including patients' perspective, voice and emotions into its scope, medicine takes a step away from biomedical paternalistic model towards more egalitarian notions of medical profession and principles of doctor–patient interaction. The relationships between doctors and patients are changing as patients get more recognition, resources and power. The asymmetry of power and knowledge in doctor–patient relations still persists, but the healthcare systems are changing. Moreover, within the context of these changes, doctors become the ones who struggle for power, authority and professional acknowledgement but, as we suppose, frequently appear to be vulnerable, lose their agency, get existentially affected, feel injustice and suffering.

We assume that social scholars pay little attention to doctors' experiences because of the binary approaches towards understanding of suffering and vulnerability: since doctors have (rather) powerful ruling position, knowledge and resources, it is patients who are perceived as powerless, vulnerable and suffering.

Nevertheless, the vulnerability of medical professionals is frequently discussed in studies dedicated to dealing with complicated clinical tasks, vulnerable groups and sensitive experiences, for instance, in the case of disciplinary processes following patients' complains [18], due to distress and professional burnout, or as a result of being traumatized due to negative patient outcomes [19]. Vulnerability of medical professionals also has class, gender and specialization dimensions. For instance, young female doctors, as well as nurses and midwives can experience more pressure due to their subordinate gendered position. Some studies show that there is a connection between the vulnerability of doctors and that of patients. Within the discipline of psychology, scholars describe the phenomenon of countertransference [20] when doctor's own problems or emotional responses are translated to patients. In the opposite direction, patients' responses and complaints can go beyond the certain situation and negatively affect the professional identity of doctors [21]. Doctors can also be seen as "second

victims" of some adverse patient events, which happened due to a medical error or to patient's condition [19,22,23].

There is evidence (mostly from psychological disciplines) that medical professionals experience psychological difficulties while providing the end of life care (especially for children), dealing with loss (for example, reproductive loss) or telling the "bad news" (e.g., [20,24]). Vulnerability of some groups of professionals depends on workload, stress and possibilities for coping with it [25,26]. However routine emotions of medical professionals and their structural reasons have gained little analytical interest within social sciences (one of the examples is [27]).

In this article we want to consider the situations, in which doctors in a Russian high-technology perinatal center become vulnerable. These vulnerabilities are hard to determine as such a priori, but they rather demand careful observation of practices and situations. We conceptualize vulnerability of doctors as associated with a lack of professional autonomy, lack of trust and authority, institutional complexity, the inconsistency of regulation and the ambiguity of rules. Vulnerabilities are expressed in "existential" and "moral" modes. The vulnerability of doctors (and other healthcare providers) usually remains invisible for both patients and public. We want to make it visible; for this, we will try to overcome the duality of the patient–doctor relationship concept and show that both sides of this interaction may be interpreted as powerful and vulnerable, and that these relationships are not binary but more complex. Power is more diffusive as determined by numerous structural limitations in concrete contexts.

Scholars of the Neo-Weberian approach in sociology of professions define professional power of a doctor as that consisting of clinical autonomy, particular knowledge and competence in medical diagnosing and curing, high social status and professionals' closure [9,28,29]. However, in different social contexts, the autonomy and powerful position of medical professionals can be challenged in multiple ways by the marketization and managerialism. In Russia, beside marketization and managerialism, we can also observe the effects of governmental paternalism [30], which systematically restricts professional power and ability to make decisions but still assigns them the main responsibility for healthcare provision. At the same time there is an extension of the scope of doctors' professional roles and obligations—they are expected to provide psychological, emotional, administrative support of patients—which they are not always able to implement. In further section we will describe the institutional context of Russian maternity care system, in which dominating managerial regulation in combination with the new market mechanisms in healthcare, considerably restrict professional power of doctors.

*3.2. Institutional Arrangement and Change of Maternity Care in Russia Causing Professional Vulnerability*

This section addresses the wider context of changing health and maternity care in post-Soviet Russia and emphasizes how changes predetermine the emergence of multiple vulnerabilities in terms of institutional complexity, the inconsistency of regulation and the ambiguity of rules. The tendency of considerable transformation of the healthcare sector and professional work in it is a world-wide phenomenon [31]. The neoliberal policy, which fosters the dominance of managerialism and market principles of regulation and financing, can be considered to be a common trend in healthcare worldwide [32] (p. 378). However, different social contexts constitute various configurations of the maternity care and challenges, shaped by neoliberal policies. That of Post-Soviet Russia, which is characterized by the quite limited professional autonomy of doctors, midwifes and nurses [33], represents the case of the appreciable challenges emerging for professional work.

In general, maternity care in Russia mostly consists of state-funded and facility-based services, which in many respects inherit the organizational arrangement and regulatory paternalistic framework from the Soviet period [34–36]. As the whole system of Soviet healthcare, maternity care used to be centrally regulated and highly standardized in terms of both the way of material provision and medical practices.

Social researches analyze health care in Soviet times and later in post-Soviet Russia as historically one of the most rigid bureaucratized systems [33,37]. Being overregulated and centralized, following the state interests and goals, the system of healthcare (and maternity care in particular) leaves little space for professional autonomy and institutionally remains insensitive to the needs and circumstances of a concrete organization, professionals and patients. We add to this investigation how some features of the institutional arrangement of maternity care in Russia set multiple vulnerability of health care practitioners.

We will analyze further how professionals became vulnerable in their routine working interactions. Our main argument is the following. Clinical power of professionals in Russian maternity care is limited not only by biomedical conditions but also by volatile non-flexible contradictory managerial-paternalist state rules and norms from one side and growth of patient demands from the other. Professionals often could not fulfill contradictory state's rules or follow consumers' numerous demands, and they became vulnerable facing moral and legal injustice from both sides—state bodies and patients. We will look shortly on legislative and institutional conditions, pronatalist state concerns, volatility and paternalism of the health care as the main structural conditions influencing on doctors' position.

Legislative contradictions can be considered one of the key features of institutional and organizational settings of health services in Russia. Perpetual change of the formal rules and regulations aggravates the conditions of systematic uncertainties. As a result, healthcare practitioners' work consists of not only professional (clinical) responsibilities and managerial tasks but also includes a lot of special structurally invisible efforts for coordination of routine activities in order to bridge institutional and organizational gaps and manage uncertainties.

Institutional conditions, which advance professionals' vulnerability, consist of the multiplicity of the controlling bodies and ongoing strengthening of the State's control over the sector of healthcare and all the activities related to childbirth. Every medical organization is an object of intent attention of the Ministry of Health, the Russian healthcare control and Russian consumer control bodies (Roszdravnadzor and Rospotrebnadzor), fire inspection, etc., and, in case of negative outcome, of the law enforcement officials.

With the statist turn in welfare policy of the Russian state [38], pronatalism has become a core part of the state's political agenda. Maternity care appears to be even more controlled and inspected sphere, as it directly relates to the National priority of demography and growth of population [30] and, hence, represents a particular concern of both the Federal and regional authorities and a particular site of control. In particular, the rates of maternal and infant mortality serve as one of the key indicators of the regional governors' performance and efficiency. Hence, each case of maternal death concerns not only medical but political agenda as well. Such state of affairs, triggered by the demographical national anxiety, also predetermines the multiplicity of the state's efforts to 'modernize' or somehow improve the system of maternity care and to make control more rigid and detailed. In practice, all these efforts comprise another set of institutional uncertainties, which enhance the professional's vulnerability.

The path of the healthcare transformation started with the Soviet collapse in 1990s, when the key trends of the reforms were the liberalization of material provision (in particular, cuts in state's expenditure on healthcare). Transformation in this period also launched the process of patients' consumerization, in particular, resulting in transformation of providers' power, authority, and domination in their relationships with patients [39]. As a result, clinics and doctors became dependent on volatile state funding and patients' pocket money.

Another unintended consequence of this perpetual institutional change is that it increased uncertainties and led to the emergence of new institutional and organizational gaps. Each of numerous reforms taken in the sphere requires adaptation to the organizational settings of the particular medical organization. The neoliberalization of the system joined with the extremely-rigid bureaucratized way of its regulation, considerably restricting the range of such adaptive strategies. For example, state orders limit both the options in medical equipment and medicines to be obtained and the procedures of procurements of the state-funded organizations (most of the maternity units in Russia). Healthcare

practitioners are to manage compensation personally (to bridge the emerging gaps) and appear to be in routine institutional uncertainty in their practical work.

Since the Soviet collapse, social processes such as the consumerization of patients' behavior [17], the commercialization of medicine [40], and the (neo)liberalization of healthcare regulation [41] have been challenging an initially paternalistic state of affairs from different angles. Patient's demand is rising for more person-centered and less medicalized approaches; care and patient-friendliness are articulated as key components of medical services, and new institutions protecting patients' interests and wellbeing are appearing. However, paternalism in doctor–patient relations and that between the state and healthcare practitioners remain an important feature of maternity care service provision, arrangement and regulation. Russian regulatory and authority bodies at various levels target the sphere of childbirth as a priority for their policies. Consequently, the state is rather reluctant to establish more egalitarian relationships between key social actors interacting in this sphere. Paternalism can thus be considered to be a core characteristic of healthcare in post-Soviet Russia, in terms of both doctor–patient interactions and relations between the state and medical practitioners as state employees.

Managerial control in combination with state paternalism frames every medical organization as the site of endless control from the side of multiple state administrative bodies with contradictory and volatile demands, who check increasing volumes of bureaucratic documentation.

In all the domains, doctor–patient relations in Russian maternity care have been transformed throughout the last two decades. In particular, consumerization of patients' behavior transforms providers' authority and domination, and maternity care remains a field of power struggle for decision-making and ability to influence care provision and organization. But at the same time, Russian childbirth services still remain a limited means of empowerment for patients and providers [39], while the state, through the increasing control and bureaucratized machinery of regulation, remains a dominant actor.

Within the last decade, we can observe a noticeable decline in trust to doctors and a growing number of those, who "find it difficult to answer" [42], which indicates the complexity and discontinuity of patient–doctor relationships. Since paternalistic model does not include much explanation and communication, patients tend to fortify their opinions and decisions with information from Internet sites, forums, blogs and channels. On the basis of this information, they can make decisions to refuse medical manipulations, vaccination, drug intake or deny the disease [43]. Besides, some medical professionals are aware of the interconnections between patients' trust and their compliance. Therefore, they are trying to implement models and protocols of communications with proven effectiveness into their practice [44].

The crisis of trust to medical professionals encourages the growth of new market segments, specialists of which pretend to have their own expertise in the field of maternity care. These include, in particular, perinatal specialists (for breastfeeding, baby sleep, baby-bearing), doulas (assistants in childbirth), specialists for postpartum recovery ("closing of birth", bath rituals, massage). In some cases, their opinion contradicts medical recommendations, which enhances distrust because, as a result, more institutionalized medical help can be interpreted by women as unnecessary and excessively medicalizing.

### 3.3. Perinatal Center in Russia as a Special Case

Since 2006 the state's investments to the sphere of healthcare in the frame of the National foreground Projects increased ('Health' initiated in 2006 and 'Modernization' in 2011–2013) and women receive a choice of maternity hospital. During the 2010s, in the frame of the 'Modernization' program, many maternity facilities have been renovated across the country, and new Perinatal Centers—the largest and the most technically advanced maternity hospitals—were constructed. However, concurrently with the statist measures, several neoliberal policies have been implemented as well, resulting in many cases in personnel and services cutbacks. In spite of the general rhetoric of the financial support, most of the healthcare organizations in Russia became a subject of so-called 'optimization' and were forced

to follow the self-maintenance logic in material provision, though still considerably restricted by the bureaucratized managerial regulation [45,46]. Therefore, position of healthcare organizations and professionals became even more unstable.

Risky cases are routed to a maternity facility equipped to assist with definite pathology, illness or complication, each of which has different equipment and personnel and provides appropriate services. The Decree № 572n, issued in 2012, specified the order of pregnant women's hospitalization, depending on the risk of complications or pathologies associated with pregnancy or childbirth [47]. As a result, since 2012 maternity care has adopted the three-level system of medical facilities, which provide different services, have different equipment and receive different financing (with a fixed price for services at each level) in accordance with their assigned status. Large maternity hospitals and perinatal centers constitute the third level of maternity care and work as medical organizations that ensure life-saving interventions for mothers and newborns. Women with high-risk pregnancies are admitted to such facilities, which are equipped with advanced technologies and highly skilled personnel.

Such a position of a perinatal center within the whole system of maternity care in Russia predetermines its organizational and institutional specificity, which in turn enhances the vulnerability of professionals working in it. The setting of a perinatal center—a particular kind of maternity facility, which deals with medical complications and pathologies—is associated with the high probability of having emotionally sensitive and even traumatic experience by pregnant women, women in labor and young parents. Such type of organizations by design accumulates the most complicated childbirth cases, and the probability of the fatal outcomes here is much higher than in any other maternity facility. As a result, it increases the emotional burden of healthcare practitioners, who inevitably deal with life and death issues.

Being the most technically developed, often the largest maternity facilities in a region, and providing multiple medical services, all perinatal centers represent a very complex organizational structure, which requires complex intraorganizational coordination and coordination with different regions of the country. Depending on the medical specialization and the presence of the research or scientific activities, perinatal centers can consist of dozens of wards and departments and hundreds of medical personnel and technical staff. In practice, this considerably increases the organization and coordination of personalized work of health practitioners and managers, sometimes, taking most of their time and attention. In addition, a perinatal center symbolically and institutionally appears to be at the cutting edge of the maternity care in Russia, and hence, is a subject of even more increased state interest and control.

New perinatal centers since 2012 deal with those cases of childbirth, which are associated with the risk of complications estimated during pregnancy. This measure implements prenatal state goals and, as statistics demonstrate, has decreased the rates of maternal and infant mortality in most of the Russian regions [48]; however, it unintentionally has led [45,46] to the centralization of maternity care and deterioration of the healthcare accessibility in regional peripheries.

## 4. Results

Our conceptual model and empirical material prove that doctors—a powerful, resourceful, agentic group—can be vulnerable and acutely aware of their helplessness when faced with the inability to save or cure a patient (or her unborn/ baby). We refer to this vulnerability as "existential." Another kind of vulnerability arises when doctors encounter "unjust" (in their terms) interpretation and evaluation of their actions. We label this vulnerability as "moral". For instance, it inductively arises when doctors are assigned responsibility for situations they could not control, have to follow contradictory regulations or get baseless complaints from patients. Both unfair claims from patients and from regulatory authorities can have legal consequences, which create symbolic and real threats.

### 4.1. Existential Vulnerability of Professionals: "There Is Something That Will Never Be Forgotten"

Existential vulnerability concerns the fact that experience related to death is "universal"—as everyone sooner or later experiences helplessness in front of death or an unbearable suffering. Nevertheless, medical professionals perform a special role in these situations, and hence, they have very specific experiences, which make them vulnerable in a special way. First, their professional role appears to be limited by the opportunities of biomedicine, which objectively cannot manage every physical condition and save every patient, but professionals tend to take such "failures" personally and emotionally hard. This is exacerbated by the fact that in reproductive medicine, death or threat of death occur to "nonconventional" demographic groups (the ones who 'should not' die)—young women and babies. Second, contemporary demographic pronatalist politic of the state concerns the increasing the birth rates and attracts a lot of attention to maternity care. As a result, every case of maternal mortality (regardless of its inevitability and numerous complications) is becoming an issue for special attention from controlling and law-enforcement bodies and a potential legal threat for all professionals who were involved in the process of treatment.

Medicine in general and midwifery and obstetrics in particular are full of situations in which a patient feels pain, suffering and fear; experiences loss or encounters negative prognosis of the treatment. Situations, in which a patient feels herself most vulnerable, include complicated clinical cases, reproductive losses, abortions for medical reasons, complications of pregnancies and births, newborn malformations and birth traumas. Medical professionals aim at saving and helping in such situations, but sometimes it goes beyond their capabilities.

Our informants have reported that they make much effort to fix any health problems they face. However, doctors, midwives and nurses still encounter situations in which there are questions of existential character and in which they feel themselves hopeless while coping with patient's death:

> "Because anyways, there are many difficult ones [clinical cases]. On a certain stage, after all, I had another sphere of medicine, I didn't lose as much as here, but here, the level of difficulty is so that loses are inevitable . . . And kind of night calls and screams . . . I mean there is something that will never be forgotten. That's when we were sitting at the department, when we were running to the resuscitation [with the baby] on our arms, you realize that the baby is terminally ill . . . That's why these are such hard, the most difficult moments" (Interview with a pediatrician)

Doctors explain to us that they will keep on trying to save the patient even in a hopeless clinical situation or in situations with negative prognoses. In cases of lethal outcome, they feel their hopelessness and this experience leaves scars for the whole life:

> "At the intern's room we find out who passed away last week. A woman, right after the operation, a severe pathology, delivery at 34th week (pregnancy was contraindicated), the baby has probably survived, there are no complaints yet. It is said that doctors from different departments rushed there and some of them were only disrupting. Note: we had planned fieldwork on that day, but we were asked not to come" (field notes, researcher's observations)

Despite the fact that the situation was rather prospective (it became clear later, during the clinical examination of the case) and was not followed by relatives' complaints or legal trial, many professionals got engaged; the case was widely discussed as stressful for the personnel. The physical condition of a woman carried fatal risks, "It was irresistible, there were no medical mistakes", (field diary, conversation with a doctor). We (as outsiders) were asked not to come to the Center for some time, presumably not due to the fatal outcome itself but due to the emotional resonance and strains of professionals.

It is important to notice that existential vulnerability arises not only in cases of lethal outcome but also in cases of negative prognosis (both for health or for life quality) and risks of lethal outcome or grievous harm. Constant encounters with complicated clinical tasks, pathologies, deaths, severe physical conditions of babies, bad prognoses unleash the process of deep reflection:



"We don't speak in a room (so that there is no noise), girls [young doctors and interns] are knitting octopuses, we speak, caress, hug, kiss. Treat babies with love. And we are very compassionate to these mothers. Pathology of nervous system is a trouble indeed. And we understand that this premature baby—we will nurse it. But what's then?" (field notes, conversation with a neonatologist)

Different wards face hard cases, death and emotions of patients to different extent. In these terms emergency room or consultative-diagnostic department would dramatically differ from resuscitation or labor wards:

"Obstetricians always fight at the forefront for life and death" (field notes, conversation with neonatologist)

"If for other departments clinical death is a stress, for us it's a job. We are the most stressed department" (field notes, conversation with intensive care nurse)

Doctors in perinatal center specialize in working with severe clinical cases; therefore, mortality, bad outcomes and poor clinical prognosis are always an inevitable part of their work. However, professionals tell about severe cases or loss with personal emotional troubles. They are worried, frustrated and it is hard for them to tolerate every case of maternal or neonatal death.

One of the emotional situations that we observed during the fieldwork was related to the potential threat for the life of a patient who refused to admit the problem and accept treatment. Professionals tell that they spent several working days on endless talks with the patient trying to convince her and one of the doctors "was so nervous that she couldn't fall asleep and was walking the streets at night" (field diary, conversation with a doctor). Professionals feel and express the existential helplessness which is accompanied by the fact that in the context of lack of trust, patients do not believe in prognosis, and doctors cannot persuade them to act in a necessary way (from their point of view).

The situation was as follows. In the hospital there was a young woman who had just given birth in another hospital and was transferred to the perinatal center for clinical reasons. Doctors believed that there was a serious threat to her life. The patient was in the intensive care unit, subjectively felt normal and insisted on discharge from the hospital. Her husband also insisted on discharge and accused doctors of overdiagnosis and forcibly keeping the woman in the hospital:

"Husband: "She was living a normal life, you found heart [problems], that's you who cannot decide, whether it is heart or kidneys . . . You make her, you forcibly hold her in the hospital . . . you can't make her do something you want. She wants to go home, she is feeling good"

Doctor: "She has a risk of death". (field notes, researcher's observations)

Professionals think that the decision of a patient is fatal—"*They make a mistake which is the size of life*" (field notes, conversation with a doctor). In this case, the doctor supposed that the patient did not realize the threat to her life despite the fact that she was given medical explanations many times. The patient and her husband relied on their previous lay experience and the experience of their social environment, interpreted the situation as an ordinary one and demanded to be discharged from the hospital. In a conversation with us, the doctor said: "We can expect nasty things, she will write to the President", i.e., there is a potential possibility of complaints and follow-up checks, especially when there is a potential threat of maternal death, each case of which is controlled by the Ministry of health and regional authorities.

As a result, patients become even more vulnerable because numerous involved professionals use "aggressive" techniques to persuade patient in order to minimize medical risks and to subordinate patient to their decision. In the described situation the doctors and the patient do not come to an agreement, and the woman refuses to continue the treatment; however, after difficult negotiations with

patients and consultations with different medical committees, professionals find a solution and transfer her to another hospital to which she agrees to go to (it is closer to home, though not specialized).

This situation is sensitive for medical practitioners not only because they can be legally prosecuted in case of death of the patient or serious harm to her, which they could predict but could not cope with, but also because they do not have enough authority and trust in the eyes of patients to protect them from lethal or disabling outcomes of clinical situations. This additional responsibility forces doctors to behave more assertively towards patients who do not believe and refuse to follow their recommendations. As a result of the lack of mutual trust, doctors are urged to use affective and "forceful" arguments, while patients respond to them with aggression and even greater distrust:

> "[Doctors] are speaking quite rough . . . It was emotionally hard for me, maybe because of the hopelessness of the situation and inability to negotiate . . . Verbally doctors are threatening and bullying her to make her stay. Although—no doubt—they make it for her benefit and may be even saving her life. [One of the doctors] doesn't sleep at night, [the other] is outlining his brutality". (field notes, researcher's observations)

At the same time, neither doctors nor nurses have professional tools and special skills for communicating sensitive topics, which at the same time is a routine for them. Neither is there a practice of calling a mediator. This often affects patients, whose emotions remain unrecognized or ignored (perceived as grotesque, or demonstrative behavior). Topics related to ethics and communication with the patient are underrepresented in the curriculums of medical schools and colleges. Psychologists, who could provide both doctors and patients with professional help, can hardly get a position in hospital because they lack legal regulations of their work and trust within medical organizations. As a result, medical personnel can usually only count on their own experiences and collective practices while discussing difficult topics with patients. Moreover, they have to direct their efforts not to emotional assistance to patients and their relatives, to colleagues or themselves, but to protecting themselves and their professional collective from subsequent sanctions connected to maternity or infant death, and then, patients suffer more as they fell themselves helpless and cheated in such kind of communication.

### 4.2. Moral Vulnerability of Professionals

Moral vulnerability emerges when professionals face unjust evaluations and critical interpretation of their actions made either by regulatory and controlling bodies (with their constantly threatening sanctions) or by patients.

### 4.2.1. "Big Brother Is Watching You"

Doctors constantly feel themselves objects of all-round control. They tell about their precarity and insecurity under controlling gaze, which is perceived as a threat to their professional status and personhood in general. Threat is a kind of "outer force" ("God forbid something happens"), which lies beyond the professional's control and creates the feeling of hopelessness:

> "I say personal insecurity when you realize that in case, God forbid, something happens, nobody will be on our side, nobody will help" (Interview with a doctor)

> "Nobody will protect doctors" (field notes), "nobody advocates for physicians in front of the public" (Interview with a pediatrician)

Doctors are meant to strictly follow the laws, recommendations, procedures and rules. As we described earlier, they have constantly been controlled by various authorities (such as SanPiN, Rospotrebnadzor, Ministry of health), which produce the rules that rapidly change and sometimes contradict each other. This is one of the consequences of ongoing reforms and hybridization of governmental paternalism and new managerialism. The legal insecurity and vulnerability are generated by multiple institutional circumstances, uncertainties and organizational gaps, which in turns are

produced by conflicting legislative requirements, organizational rigidity and material constraints that professionals are talking about (see Section 3). Professionals constantly feel their precarity in such conditions. In addition, the control over doctors is strengthened by the promotion of state demographic priorities of increasing fertility and growing attention to maternity care. Professionals say: "Big brother is watching you" (field notes). During the fieldwork, we could regularly see health practitioners discussing future inspections and dangers they can possibly bring:

> "Fines are inevitable. [The nurse] believes that they just have to reconcile with it. The only question is about the size and the legal subject—a (physical) person or a corporate body (organization). Sometimes it is easier just to put the responsibility on oneself than to arrange an administrative commission". (field notes)

> "I ask her [the nurse] why is this so bad (about administrative commission). Is it because there are so many violations or because they cannot be fixed? She says yes, there are too many inconsistencies, which she (and nobody) doesn't know how to fix for the period of inspection. "My fantasy is not enough to pull the wool over inspectors' eyes! (she means—how to represent themselves in the best way for the inspection"". (field notes)

Our data supports the claim that formal requirements are often contradictory and cannot be met in full due to circumstances which are beyond professionals' control. In emic terms, the phrase of the doctor would be "*the chaos is everywhere within the medicine*" (field notes). Professionals act in patients' interests and cope with gaps in their professional daily routine by frequently breaking certain formal rules and recommendations. Consequently, they can potentially be accused or sanctioned. Professionals clearly understand it and say with irony that: "my task is to prepare everything for the prosecutor so that he can't get to me" (field notes).

Take the example of solving a problem of insufficiency of medications and equipment, which is derived from the organizational inability to buy them quickly. The doctors can face the two options: not to follow clinical recommendations and cure the patient with available treatment or search for the prescribed recommended medication by using informal instruments. For instance, professionals sometimes make purchases themselves, which is considered illegal:

> "Nurses buy containers and special tools with their money. This weekend they plan to go shopping together" (field notes)

> "They [parents] bring [money] to the discharge—doctors leave it in the department for medications. [My relative] brings suitcases of a foreign medicament. Resuscitation [department] also brings it from vacation. Sometimes we buy it ourselves" (field notes, conversation with a doctor)

> "They borrow [medication from other departments], but this is a serious violation of rules" (field notes)

Professionals are vulnerable also due to the risk of detention for informal payments, which are explained by low wages and a necessity to survive: "There is informal money, and that's life. And so how could one live on these wages, when you need to feed the family" (field notes, conversation with a doctor). This is a hidden topic which is ambivalently evaluated in medical community (about informal payments see [39]).

Moral panics in media incite mistrust and aggression towards medical professionals. Cases of infant and maternity death, birth traumas and various iatrogenic conditions regularly become a topic for massive public debates. All together, the increased attention of the Investigating Committee, media coverage and institutional controversies comprise the particular settings, which stimulate patients' complaints and invent new forms of control but leave little opportunity for medical professionals to deal with it. The control becomes more pervasive due to new instruments, such as audio- and

video-recordings of sessions with patients, online sites for commenting on and evaluating doctors and medical organizations, professional associations aimed at representing the interests of patients (League of Protectors of Patients, Investigation Committee). At the same time, medical professionals lack resources and social and professional support, to protect themselves in situations of legal prosecution or media scandals, which makes them feel constantly vulnerable. On the one hand, patients try to get a voice and empowerment, which were unachievable within the paternalistic model. On the other hand, mistrust makes them more demanding and blocks the possibilities for dialog, cooperation and compliance. Some patients are conscious that doctors and medical organizations are very sensitive to complaints and therefore try to get profit during the process of cure (extra services or financial compensations). This practice was reflected in terms used in medical environment—"the patients' terror" and "an extremist patient".

### 4.2.2. "An Extremist Patient"

Another type of injustice and vulnerability is related to the rise of complains and grievances of patients, many of which are deemed as unfair by physicians. Professionals take complaints very hard as they can lead to administrative and material sanctions. Patients are becoming more demanding in their ethics and style of communication and self-sufficient explanations. The principle "Doctor knows best" does not work universally any more. Patients are trying to get more control over the situation, evaluate doctors and hospitals, describe their experience, write down comments on the Internet. Patients are becoming more exacting as consumers [39].

For medical professionals in Russia this is a relatively new situation, and they often feel themselves helpless victims of unrealizable demands and injustice and unready to solve the problem. They distinguish a certain type of patient, which represents a threat—these are "aggressors" or "extremists". They write complaints to different controlling bodies and online sites. According to professionals, they act aggressively, behave unethically, make unrealizable demands and "biased" complaints:

"Oh, mother, within three days she managed to write eight complaints to all instances of the world! Listen, we ... we are absolutely unprotected from this. A person can write anything: a positive feedback, a negative feedback. I like—I didn't. Absolutely biasedly" (Interview with an administrator)

"The doctor says: a mom was brawling (today) because she didn't get the medication. It costs 16,000 rubles; we ordered it; it will be delivered (in a few days). But she wants to get discharged on Saturday, because of the birthday. She says: "Take it wherever you want, at least buy it and pay it yourself"". (field notes)

Complaints lead to reputational loses and emotional costs. We were told about a complaint, which was considered unsubstantiated. The doctor, who was mentioned in the complaint, was taking the situation very hard and was even about to quit the job:

"There were two proceedings. The doctor had been going crazy all five days before that. She was sending messages to me: "Maybe I should quit my job?" ... Reputationally this is very painful ... not to crush this person". (Interview with an administrator)

Complaints can also be made on the basis of communicational and service problems. The doctor tells about a complainant who considers,

"The childbirth went well, thanks to your specialists". And then, somebody didn't open the door in a right way, somebody offered something wrong, something that made them indignant and provoked to [write down] two pages. They didn't like the magnet key (for exit) for some reason; I mean, and so on ... You were not served? What you were not served? In what way you were not served? ... Do you understand that all this, in truth, deeply hurts medical practitioners". (Interview with an administrator)

Hospital meal, late discharge, intrusive photographers in a check-out room and other reasons which lie beyond the responsibilities of a health practitioner, can become a basis for a complaint. The aim of "patients-aggressors", who are selfish as considered by professionals, is to get financial profit or moral satisfaction.

Doctors are in a situation where they are becoming more controlled by the patients; they can be complained about every single moment. Every patient can record a conversation and post it on the Internet: "Patients are taking pictures of us with their mobile phones, and we feel and consider this" (field notes, conversation with a nurse).

According to our data, lack of trust and absence of compliance become a background for blaming physicians for negligence, disregard or dishonesty. During a fieldwork, we repeatedly observed how hard it can be for doctors to conduct a dialog with patients, especially those in a critical or threatening situation. Doctors who are striving to solve difficult clinical tasks describe their job as physically hard and emotionally charged, frequently telling about emotional burnout. Patients often do not appreciate their efforts—they do not see and cannot evaluate the complexity of this work under the conditions of institutional contradictions and multiple all-round control. Patients, who are physically and emotionally vulnerable themselves, are suffering of neglect, discomfort, and misunderstanding.

As a result, a lot of (potential and real) situations of discontent and complaints are based on a conviction that the doctor is dishonest and acts in his or her own interests. Patients tend to see deception when the actions and interpretations of doctors remain unclear, confusing and contradictory to their own life experience.

Therefore, doctors, whose social position is provided with power, resources and competence, in some cases appear to be vulnerable both in terms of existential events, which are out of their control, and in terms of unjust evaluations of their actions and sanctions against them; their power and resources appear to be insufficient. Vulnerability of professionals remains invisible as it does not correspond with their social position. However, it negatively affects the patients. For a doctor who is herself hardly struggling with existential situations and threatening sanctions, it is difficult to provide sufficient support to suffering patients or their relatives. A doctor who does not have the opportunity to act in the best interests of a patient or has to break the law in order to do so can only aggravate the vulnerable position of a patient. Therefore, as a result of doctors' vulnerability which is related to institutional and organizational contexts, patients become even more vulnerable.

## 5. Discussion and Conclusions

This article contributes into the contemporary discussion on vulnerability of medical professionals. We are reacting on two trends in literature on vulnerability. The first one focuses on deprivation, marginalization, disadvantage, poverty and social problems [49]. Doctors cannot be attributed to this group. Another trend considers stress and burnout of professionals, but ignores structural and contextual basis for their vulnerability. Our research aims at filling in this gap.

We set the task to examine an invisible vulnerability of a group, which is considered as powerful and resourceful, doctors in Post-Soviet context, in a special site, perinatal center. As a rule, vulnerability is attributed to patients (especially such as terminally and mentally ill) as passive and not enough knowledgeable help recipients. In spite of the politics of neoliberal choice and empowerment of patients', their agency and resources are restricted; they experience bodily and emotional suffering. Within binary approaches to the understanding of power relations, doctors are opposed to patients—they have power, agency, they are not supposed to suffer and are not considered vulnerable. We critically refer to this point.

Our first conclusion refers to the subject of vulnerability. It is methodologically important not to define certain groups as (not) vulnerable by default. Such artificial narrowing of the field of analysis might derive into disregard for "unexpected" forms of vulnerability. We have to be sensitive to practices, interactions and emotional displays of all the participants, not only the ones who are determined as a priori less powerful. During an ethnographic fieldwork we discovered multiple vulnerabilities whose

boundaries are transparent. Doctors are conventionally perceived as powerful and affectively neutral, but in a number of situations, they lose power, cannot manifest their agency and face lack of resources. Their sufferings (existential and moral), as a rule, are invisible, denied and ignored.

Our second conclusion relates to the meaning of context in exploring vulnerability and vulnerable groups. In a context of hybridization of paternalism, managerialism and marketization of Russian healthcare, doctors feel the injustice of increasing and constantly changing requirements from different instances, which cannot be simultaneously met as they contradict each other. Doctors' autonomy is restricted, their actions are regulated by multiple and frequently contradicting rules; one can hardly influence one's own working conditions or choose optimal treatment strategies for patients. As a result, doctors turn out to be not only existentially vulnerable but they feel themselves legally insecure and experience injustice—that is moral vulnerability.

Our third conclusion is that patients who a priori can be vulnerable, in certain conditions might suffer even more because of (subjective) insufficiency of care and lack of attention from vulnerable doctors. Due to the low level of mutual trust, some patients ("extremists" in emic terms) accuse doctors of deception or neglecting their interests. Doctors consider such complaints unjust. They make more efforts to protect themselves from sanctions than to support their patients.

Our last conclusion concerns the position and actions of the researcher in the empirical field when dealing with multiple vulnerabilities. We recognize this position as complex and ambivalent. On the one hand, the vulnerability of informants is associated with sensitivity, which they do not want to show, and it can be emotionally difficult or dangerous to openly discuss it. Or just the opposite, the stories and the situation become emotionally oversaturated. Moreover, the researcher experiences emotional difficulties during such conversations or observations, which are not always easy to cope with. Cases of existential vulnerability also create additional challenges and limitations for accessing the field and collecting empirical data—in an emotionally overcharged situation, a sociologist in the field as an outsider creates extra burden for participants, so she probably will be excluded from the most problematic situations.

**Author Contributions:** Conceptualization, D.L., A.N. and A.T.; methodology, D.L., A.N. and A.T.; analysis, D.L., A.N. and A.T.; investigation, D.L., A.N. and A.T.; data curation, D.L. and A.N.; writing—original draft preparation, D.L., A.N. and A.T.; writing—review and editing, D.L., A.N. and A.T.; supervision, A.T.; project administration, D.L.; funding acquisition, D.L., A.N. and A.T. All authors have read and agreed to the published version of the manuscript.

**Funding:** The research was funded by the Russian Science Foundation (Project No 19-78-10128).

**Acknowledgments:** We would like to acknowledge health care practitioners of the perinatal center for their organizational support and their time spent on the participation in the research. We also would like to thank our colleague, the head of the project funded by RSF, Ekaterina Borozdina, for the financial support of the research.

**Conflicts of Interest:** The authors declare no conflict of interest.

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
