# Peer review of "Multiple Vulnerabilities in Medical Settings: Invisible Suffering of Doctors"

_societies, doi:10.3390/soc10010005_

Round 1

Reviewer 1 Report

The subject is extremely interesting and painful for medical community. The article is a little to long. The chosen topic is of scientific interest but should devote more attention in exposing the arguments and presenting the information.

This manuscript may be improved if the author addresses the following comments.

The paragraph between 538-545 it's not clear need to be rewrite

The conclusion need to clear short and specific.

Please recheck the References order.

Please double check the article by a native english reader.

Thanks for the opportunity of reading the article.

Reviewer 2 Report

Review of “Multiple vulnerabilities in medical settings: invisible suffering of doctors”

The paper investigates the vulnerability of doctors and discuss its implication for the health system and in particular for the patients. The manuscript also describe the specific situation/features of post-Soviet healthcare systems in Russia where the “case study” was carried out.

The paper face an interesting argument and can contribute to analyze this relevant problematic. However, in my opinion, the paper have some relevant weaknesses and should be improved before the publication.

Hereafter, I try to provide a list of the main weakness points highlighted in the review:

The introduction is very brief and it should be extended. In particular, it lacks the explicit aim of the research and the justification of the research. I think this aspect is very important because a clear justification can provide the reason why it is important scientifically this research. This is a very important point. The article does not have a background section, although some past evidence about vulnerabilities in healthcare (and some potential causes) are reported in section 3 (Results). I think the paper should include an appropriate section of background. This section can/should include the content of section 3.1 and 3.2, but it should also include more healthcare literature. Hereinafter, some useful references are provided: Stefanini, A., Aloini, D., Benevento, E., Dulmin, R., & Mininno, V. (2019). A data-driven methodology for supporting resource planning of health services. Socio-Economic Planning Sciences, 100744. https://doi.org/10.1016/j.seps.2019.100744.; - Sandars, J., and Heller, R. (2006), “Improving the implementation of evidence‐based practice: a knowledge management perspective”, Journal of evaluation in clinical practice, Vol. 12 No. 3, pp. 341-346.; - Wheelock, A., Suliman, A., Wharton, R., Babu, E. D., Hull, L., Vincent, C., ... & Arora, S. 2015. The impact of operating room distractions on stress, workload, and teamwork. Annals of surgery, 261: 1079-1084.; - Arora, S., Sevdalis, N., Nestel, D., Woloshynowych, M., Darzi, A., & Kneebone, R. 2010. The impact of stress on surgical performance: a systematic review of the literature. Surgery, 147:318-330.; - Dahl, A. B., Abdallah, A. B., Maniar, H., Avidan, M. S., Bollini, M. L., Patterson, G. A., ... & Ridley, C. H. 2017. Building a collaborative culture in cardiothoracic operating rooms: pre and postintervention study protocol for evaluation of the implementation of teamSTEPPS training and the impact on perceived psychological safety. BMJ open, 7: e017389.; - Stefanini, A., Aloini, D., Dulmin, R., Mininno, V.; Linking Diagnostic-Related Groups (DRGs) to their processes by process mining (2016) HEALTHINF 2016 - 9th International Conference on Health Informatics, Proceedings; Part of 9th International Joint Conference on Biomedical Engineering Systems and Technologies, BIOSTEC 2016, pp. 438-443.; - Singer, S. J., Molina, G., Li, Z., Jiang, W., Nurudeen, S., Kite, J. G., ... & Berry, W. R. 2016. Relationship between operating room teamwork, contextual factors, and safety checklist performance. Journal of the American College of Surgeons, 223: 568-580. The method of the research seems appropriate, but the “second part”, the “triangulation with data gathered in the past” of the methodology should be better clarified. The sentence “The results of the current research have also been triangulated with the previously gathered data, which included:…” and the list below deserve some more explanations (How do you use such data?) Finally, I remark this minor problem. At row 80-81, this sentence should be modified: “In a text we frequently use the term “professionals”, which we define as medical practitioners of different departments and specializations.”

Round 2

Reviewer 2 Report

The authors have done a good effort in improving their paper. The introduction and background section were improved and appear now appropriate.

However, my doubt remain about the discussion and conclusion section.

I think is the most relevant point remaining. Particularly, I think you should discuss your findings in light of results obtained by similar/related studies. The discussion of your results is very important because it provide "value" to your research and permit a comparison with past literature. I think this aspect is really important because otherwise the reader can not understand the value of your findings respect to previous ones.

In addition, I think you should add also some more descriptions about the practical implications of your research (under the doctor and health managers "light").